# Spatial distribution and characteristics of women reporting cervical cancer screening in Malawi: An analysis of the 2020 to 2021 Malawi Population-based HIV Impact Assessment survey data

Hussein Hassan Twabi[1,2]*, Takondwa Charles Msosa[1,3], Samuel James Meja[1], Madalo Mukoka[1,4], Robina Semphere[1,5], Geoffrey Chipungu[1], David Lissauer[2,6], Maria Lisa Odland[2,6,7], Jenny Tudor[2], Chisomo Msefula[1], Marriott Nliwasa[1]

1 Kamuzu University of Health Sciences, Blantyre, Malawi, 2 Institute of Life Course and Medical Sciences, University of Liverpool, Liverpool, United Kingdom, 3 Department of Global Health, Amsterdam University Medical Centres, University of Amsterdam, Amsterdam, the Netherlands, 4 London School of Hygiene and Tropical Medicine, London, United Kingdom, 5 University of Glasgow, Scotland, United Kingdom, 6 Malawi-Liverpool-Wellcome Trust Programme, Blantyre, Malawi, 7 Department of Obstetrics and Gynaecology, St. Olavs University Hospital, Trondheim, Norway

* husseintwabi@hotmail.com

**Data Availability Statement:** Data cannot be shared publicly because of the data is owned by

## Abstract

### Background

Malawi has one of the highest incidence and mortality rates of cervical cancer in the world. Despite a national strategic plan and the roll-out of VIA and screen-and-treat services, cervical cancer screening coverage in Malawi remains far below the national target.Using a nationally representative sample of women enumerated in the Malawi Population-based Impact Assessment (MPHIA) survey we estimated the prevalence and spatial distribution of self-reported cervical cancer screening as a proxy for uptake in Malawi.

### Methods

MPHIA was a nationally representative household survey in Malawi, targeting adults aged 15 and above, that employed a cross-sectional, two-stage, cluster design. The primary aim of MPHIA was to assess the regional prevalence of viral load suppression and the progress towards achieving the UNAIDS 95-95-95 goals among adults aged 15 and above. The survey was carried out between January 2020 and April 2021. Prevalence of self-reported cervical cancer screening by different characteristics was estimated accounting for the survey design using the Taylor series approach. We used univariable and multivariable logistic regression approaches to examine associations between the prevalence of cervical cancer screening and demographic characteristics.

the following institutions: the Malawi Ministry of Health (MOH), the Malawi National AIDS Commission (NAC), The United States (US) President's Emergency Plan for AIDS Relief (PEPFAR), the Centers for Disease Control and Prevention (CDC), Malawi, Centers for Disease Control and Prevention (CDC), Atlanta, the Individual Career and Academic Plan (ICAP) at Columbia University, New York, and ICAP at Columbia University, Malawi. The authors were only given permission to use the data for the purposes of the request submitted to the CDC and MOH via the ICAP Columbia website. The authors are thus legally bound by the agreement with the aforementioned institutions which outlines the following: MPHIA 2020-2021 data can be requested for use in research and analysis under the following conditions: • Recipient will use this data only for the purpose of the research and analysis described in this data request. The recipient will submit a new request if they intend to use the data for another purpose. • Recipient will not share this data with other researchers, with the exception of those listed in this data request as co-researchers for the project. • Recipient will ensure that co-researchers are aware of and follow the terms of this data use agreement. • Recipient will treat all data as confidential. Recipient will not use the data to deliberately compromise or otherwise infringe on the anonymity of participants' information and their right to privacy and will not attempt to identify any individual, household, or community in the survey based upon these data. • Recipient will not publish any result in which participants, EAs or communities can be identified. • Recipient will keep data in a secure location where it cannot be accessed by unauthorized users. • Recipient will not use this data for any commercial venture. • Recipient agrees that this agreement terminates immediately upon any breach by the recipient of the data or any co-researchers. Data are available from the ICAP Columbia Institutional Data Access site for researchers who meet the criteria for access to confidential data. The data underlying the results presented in the study can be accessed by placing a request on the PHIA ICAP Columbia website: https://phia-data.icap.columbia.edu/datasets?country_id=3 Additionally, the ICAP at Columbia University may be contacted via the following mailing address: ICAP at Columbia University, 722 West 168th Street, New York, NY 10032

**Funding:** The author(s) received no specific funding for this work.

**Competing interests:** The authors have declared that no competing interests exist.

## Findings

A total of 13,067 adult (15 years and older) female individuals were surveyed during the MPHIA 2020 to 2021 survey, corresponding to a weighted total of 5,604,578. The prevalence of self-reported cervical cancer screening was 16.5% (95% CI 15.5–18.0%), with women living with HIV having a higher prevalence of 37.8% (95% CI 34.8–40.9) compared to 14.0% (95% CI 13.0–15.0) in HIV negative women. The highest prevalence of screening was reported in the Southwest zone (SWZ) (24.1%, 95% CI 21.3–26.9) and in major cities of Blantyre (25.9%, 95% CI 22.9–29.0), and Lilongwe (19.6%, 95% CI 18.0–21.3). Higher self-reported screening was observed in women who resided in urban regions ((22.7%; 95% CI 21.4–24.0) versus women who resided in rural areas (15.2%; 95% CI 14.0–16.8). Cervical cancer screening was strongly associated with being HIV positive (aOR 2.83; 95% CI 2.29–3.50), ever having been pregnant (aOR 1.93; 95% CI 1.19–3.14), attaining higher education level than secondary education (aOR 2.74; 95% CI 1.67–4.52) and being in the highest wealth quintile (aOR 2.86; 95% CI 2.01–4.08).

## Interpretation

The coverage of cervical cancer screening in Malawi remains low and unequal by region and wealth/education class. Current screening efforts are largely being focussed on women accessing HIV services. There is need for deliberate interventions to upscale cervical cancer screening in both HIV negative women and women living with HIV.

## Background

Cervical cancer is one of the commonest cancers diagnosed in women and is the fourth leading cause of cancer deaths in women globally [1]. Cervical cancer is the leading cause of cancer-related death in low- and middle-income countries, with the highest regional incidence and mortality occurring in sub-Saharan Africa [1].

In 2020, there was an estimated 604,000 new cases of cervical cancer and 342,000 cervical cancer-related deaths globally, representing age-standardised incidence and mortality rates of 13.3 per 100,000 and 7.3 per 100,000 respectively [1]. The age-standardised incidence rate of cervical cancer in Southern Africa in 2020 was 36.4 per 100,000, three times the global incidence rate, with an age-standardised mortality rate of 20.6 per 100,000 [1].

Malawi is a country with one of the highest incidence and mortality rates due to cervical cancer in the world, registering twice the age-standardised incidence and mortality rates of Southern Africa—age-standardised incidence rate of 67.9 per 100,000 and age-standardised mortality rate of 51.5 per 100,000 [1]. This represents an increasing trend in the occurrence of new cases of cervical cancer in the region [2]. As an AIDS-defining condition, the risk of cervical cancer is significant in the region, owing to the high HIV prevalence irrespective of antiretroviral therapy (ART) uptake [3–5].

Cervical cancer is considered to be one of the most preventable and treatable malignant conditions worldwide, [3] yet efforts to effectively screen and manage this disease fall short. The Malawian screening programme largely relies on the screen-and-treat initiative recommended by the Malawi Ministry of Health (MoH) through Sexual and Reproductive Health Directorate, which utilises visual inspection with acetic acid (VIA) and cryotherapy [6].

The World Health Organisation (WHO) defines cervical cancer elimination as reducing and maintaining the incidence of cervical cancer to less than four in 100,000 women [7]. With this in mind, in May 2018, the WHO issued a call to eliminate cervical cancer as a public health problem, which proposed a new 90-70-90 target for elimination of cervical cancer– 90% HPV vaccination coverage, 70% screening coverage, and 90% of cervical lesions treated by 2030 [7]. The Malawi Cervical Cancer Control Programme (CECAP) outlines a strategic plan to eliminate cervical cancer which proposes that at least 80% of women aged between 25 and 49 years should have been screened for cervical cancer using visual inspection methods for the first time within a 12-month period [8]. This is a higher target that includes a younger age group than the 70% by 35 years prescribed by the WHO in its global strategy of cervical cancer elimination [9].

Despite this strategy, as well as a national roll-out of VIA and screen-and-treat services in Malawi, coverage still remains far below the national target, with a high loss-to-follow-up of previously screened women [6]. By 2015, only 26.5% of women were reported to have screened for cervical cancer, with less than half (43.3%) of those with cryotherapy eligible VIA positivity actually receiving the cryotherapy [6]. An ecological study by Ng'ambi et al in 2023 reported up to 78% cervical cancer screening coverage among HIV positive Malawian women in 2021, an increase of 24% from the 54% reported in 2019 [10]. However, it is important to note that the data utilised in this study were derived from PEPFAR programmatic databases, and thus, the near-target screening uptake observed can be generalised to facilities with PEPFAR support only.

Since the study by Msyamboza et al. in 2016, [6] there has been no updated countrywide estimate of the cervical cancer screening uptake by the general population in Malawi. The Malawi Population-based HIV Impact Assessment (MPHIA) collected data on the self-reported screening practices of a sample of both women living with HIV and HIV negative women representative of the entire country. We thus aimed to estimate the prevalence and spatial distribution of self-reported cervical cancer screening in Malawi from a representative sample of women enumerated in the MPHIA survey, and to understand the factors associated with cervical cancer screening amongst the survey respondents.

## Methods

### Study design, setting and participants

This was a cross-sectional analysis of data from the Malawi Population-based Impact Assessment (MPHIA) conducted in 2020 to 2021. MPHIA was a nationally representative household survey in Malawi, targeting adults aged 15 and above, that employed a cross-sectional, two-stage, cluster design. The primary aim of MPHIA was to assess the regional prevalence of viral load suppression and the progress towards achieving the UNAIDS 95-95-95 goals among adults aged 15 and above. The survey was conducted between January 2020 and April 2021, and was a collaborative effort between the Malawi Ministry of Health and the International Centre for AIDS Care and Treatment Programs (ICAP) at Columbia University, funded by the President's Emergency Plan for AIDS Relief (PEPFAR), with technical support from the US Centres for Disease Control and Prevention (CDC). Details of the MPHIA sampling frame, study design, and study procedures are available in the full report [11].

### Data collection

Interviews were conducted by trained interviewers using mobile tablets equipped with the Census and Survey Processing System (CSPro) software to administer household and individual questionnaires to participants. The individual interviews included demographics, sexual

behaviours, participants' history of HIV testing and treatment, and the history of accessing cervical cancer screening.

## Variable definitions

The dependent variable in this analysis was self-reported cervical cancer screening. This was a dichotomous variable obtained directly from the response to the survey questionnaire question of whether a woman had ever been screened for cervical cancer.

We included the following independent variables in our analysis: Age, HIV status, History of Pregnancy, Geographical Zone, Residence Type, Education, Occupation, Wealth Quintile, Marital Status, and Access to Modern Contraceptive Methods. Age was categorised based on the target age groups outlined in the National Cervical Cancer Control Strategy, where the target demographic for cervical cancer screening are women between the ages of 25 and 49 years [8] As such, three ordinal classes were derived as follows: age of less than 25 years, age of 25 to 49 years, and age of 50 years and older. Age categorisations were also made based on the WHO cervical cancer elimination strategy targets, which included age of less than 35 years, age of 35 to 45 years, and age of 46 years and older.

Geographical zones were defined by pre-existing administrative zones utilised by the Malawi Ministry of Health in the surveillance of diseases in the country–S1 Fig. These zones include the Southeast, Southwest, Central East, Central West, and the Northern zone. Our analysis included Blantyre City and Lilongwe City as separate areas, as these are important in the surveillance of the HIV epidemic in the country and were thus separated in the MPHIA survey. Access to modern contraceptives was defined by whether a respondent reported using one of the following: female sterilisation, the pill, intrauterine devices, injectable contraceptives, implants, or condoms. All other independent variables were classified based on the categorisations defined by the MPHIA survey–S1 Table.

## Statistical analysis

Statistical analyses were conducted in R statistical computing software (R version 4.3.0 (2023-04-21 –R Core Team (2023), https://www.R-project.org/)). The survey design was accounted for using the Taylor Series Variance Estimation. Participant characteristics were described using median and interquartile range (IQR) for continuous data, and frequency and proportions for categorical data. Baseline characteristics were compared between those who reported having undergone cervical cancer screening and those who reported never having undergone cervical cancer screening using Chi-square tests and Wilcoxon Rank Sum tests.

Univariable and multivariable logistic regression models were used to obtain odds ratios (ORs) and 95% confidence intervals (CIs) to demonstrate the association between each independent variable and the outcome variable. Variance inflation factors (VIF) were estimated to examine multicollinearity between the independent variables in a regression model. Variables with a variance inflation factor of more than 10 were considered highly multicollinear and were removed one at a time to determine the final model–S2 Table.

## Ethical considerations

The MPHIA study protocol was reviewed and approved by the Institutional Review Boards of the CDC, Columbia University, and the National Health Research Ethics Committee in Malawi. Adult (18 years and older) participants provided written informed consent to participate in the survey. Participants aged less than 18 years provided assent and consent was obtained from their legal guardians. The respondents were recruited only when they consented

to participate in the study. During the consent process, respondents were told in advance that the data would be used in future research.

For this study, a request for access to data was made to the ICAP at Columbia University through their public access PHIA website (https://phia-data.icap.columbia.edu/datasets?country_id=3). Access was granted for fully anonymised data, which was used for this study.

## Results

### Study participants

A total of 12,815 households and 26,519 participants who were 15 years and older were surveyed during the MPHIA survey. There were 22,662 participants who had undergone HIV testing and had HIV test results. Of these, 13,067 were female and were included in the analysis, corresponding to a weighted total of 5,604,578.

### Characteristics of survey participants

Table 1 describes the baseline characteristics of the survey participants included in the analysis. The median age of the participants was 30.0 (IQR 21.0, 42.6), with 47.3% (2,650,403/5,604,578) being within the target age group (25 to 49 years) for cervical cancer screening. The proportion of women who were HIV positive was 10.5% (587,488/5,604,578), of whom 88.8% were on ART (521,529/587,488). The majority of women (82.0%, 4,587,368/5,604,578) reported to have ever been pregnant before, and were mostly married or living together with a partner (88.1%, 3,246,174/5,604,578).

The women surveyed were mostly from rural areas (82.6%, 4,629,194/5,604,578), with more than two thirds of the population coming from the Central West (20.8%), the Southeast (20.6%) and the Southwest (20.5%) zones. The North, Central East, Lilongwe City and Blantyre City only accounted for 11.3%, 15.8%, 5.9% and 5.2% respectively.

The commonest education level was primary school level (64.0%, 3,582,010/5,604,578), with most women reporting to be unemployed (64.6%, 1,248,132/1,932,701). Survey respondents were largely from the middle to highest wealth quintiles (20.7%, 20.8% and 20.4% for the middle, fourth and highest quintiles respectively).

Participants reporting having undergone cervical cancer screening were significantly older than those who had never undergone cervical cancer screening (median age of 36.0 [28.0, 45.0] versus 28.0 [20.0, 41.0], p <0.001). The target age group (25 to 49 years) outlined in the National Cervical Cancer Control Strategy was most represented in those who reported to have screened than in those who had never screened (632,099/925,235 [68%] versus 2,018,304/4,679,343 [43%], p <0.001). Women who reported having undergone cervical cancer screening had a higher proportion of HIV positivity (24.0% versus 7.8%, p <0.001) than in those who had never been screened and had a higher proportion that had ever been pregnant (96.2% versus 79.2%, p <0.001).

### Prevalence and geographical distribution of self-reported cervical cancer screening

The overall prevalence of self-reported cervical cancer screening was 16.5% (95% confidence interval (CI) of 15.5 to 18.0%). Among women living with HIV, the prevalence was 37.8% (95% CI 34.8–40.9), compared to 14.0% (95% CI 13.0–15.0) in HIV-negative women.

Fig 1A illustrates the geographical distribution of cervical cancer screening in the country. The highest prevalence of screening was reported in Blantyre City (25.9%, 95% CI 22.9–29.0), followed by the Southwest (24.1%, 95% CI 21.3–26.9) and Lilongwe City (19.6%, 95% CI 18.0–

**Table 1. Weighted summaries of the characteristics of survey participants.**

| Characteristic | Overall, N = 13,067 | Screened for cervical cancer | | |
| --- | --- | --- | --- | --- |
| | | No, N = 10,685 (84.5%) | Yes, N = 2,382 (16.5%) | P values |
| **Median age in years (IQR)** | 30.0 (21.0, 42.6) | 28.0 (20.0, 41.0) | 36.0 (28.0, 45.0) | <0.001 |
| **Age group based on CECAP targets (%)** | | | | <0.001 |
| Less than 25 years | 4,207 (35.7%) | 3,888 (40.0%) | 319 (13.8%) | |
| 25 to 49 years | 6,633 (47.3%) | 4,978 (43.1%) | 1,655 (68.3%) | |
| 50 years and older | 2,227 (17.0%) | 1,819 (16.9%) | 408 (17.8%) | |
| **Age group based on WHO targets (%)** | | | | <0.001 |
| Less than 35 years | 7,494 (60.8%) | 6,482 (63.9%) | 1,012 (45.1%) | |
| 35 to 45 years | 2,743 (18.1%) | 1,953 (15.7%) | 790 (30.2%) | |
| 46 years and older | 2,830 (21.1%) | 2,250 (20.4%) | 580 (24.6%) | |
| **HIV status (%)** | | | | <0.001 |
| Negative | 11,371 (89.5%) | 9,657 (92.2%) | 1,714 (76.0%) | |
| Positive | 1,696 (10.5%) | 1,028 (7.8%) | 668 (24.0%) | |
| *HIV positive on ART (%)* | 1,522/1,696 (88.8%) | 882/1,028 (84.9%) | 640/668 (95.1%) | <0.001 |
| **Ever been pregnant (%)** | | | | <0.001 |
| No | 1,874 (18.0%) | 1,792 (20.8%) | 82 (3.8%) | |
| Yes | 11,166 (82.0%) | 8,866 (79.2%) | 2,300 (96.2%) | |
| Missing | 27 | 27 | 0 | |
| **Geographical zone (%)** | | | | <0.001 |
| North | 1,363 (11.3%) | 1,124 (11.3%) | 239 (11.3%) | |
| Central East | 2,090 (15.8%) | 1,882 (17.2%) | 208 (9.0%) | |
| Central West | 1,727 (20.8%) | 1,519 (21.9%) | 208 (14.8%) | |
| Lilongwe City | 741 (5.9%) | 584 (5.7%) | 157 (7.0%) | |
| Southeast | 3,315 (20.6%) | 2,756 (20.8%) | 559 (20.0%) | |
| Southwest | 3,120 (20.5%) | 2,306 (18.6%) | 814 (29.9%) | |
| Blantyre City | 711 (5.2%) | 514 (4.6%) | 197 (8.1%) | |
| **Residence type (%)** | | | | <0.001 |
| Urban | 2,295 (17.4%) | 1,735 (16.1%) | 560 (23.9%) | |
| Rural | 10,772 (82.6%) | 8,950 (83.9%) | 1,822 (76.1%) | |
| **Education level (%)** | | | | <0.001 |
| No education | 1,796 (13.6%) | 1,552 (14.3%) | 244 (10.2%) | |
| Primary | 8,377 (64.0%) | 6,886 (64.3%) | 1,491 (62.4%) | |
| Secondary | 2,595 (20.2%) | 2,038 (19.6%) | 557 (23.3%) | |
| More than secondary | 279 (2.2%) | 189 (1.8%) | 90 (4.1%) | |
| Missing | 20 | 20 | 0 | |
| **Occupation (w%)** | | | | <0.001 |
| Unemployed | 3,009 (64.6%) | 2,490 (67.8%) | 519 (51.7%) | |
| Mining | 13 (0.3%) | 11 (0.3%) | 2 (0.2%) | |
| Transport | 12 (0.2%) | 8 (0.2%) | 4 (0.4%) | |
| Construction | 52 (1.1%) | 36 (0.9%) | 16 (1.6%) | |
| Uniformed personnel | 44 (1.0%) | 25 (0.7%) | 19 (2.2%) | |
| Informal trade | 827 (16.9%) | 567 (14.9%) | 260 (25.1%) | |
| Garment industries | 21 (0.4%) | 14 (0.3%) | 7 (0.6%) | |
| Housekeeper | 260 (5.9%) | 198 (5.8%) | 62 (6.3%) | |
| Student | 64 (1.8%) | 61 (2.2%) | 3 (0.4%) | |
| Other | 375 (7.8%) | 267 (6.9%) | 108 (11.4%) | |
| Missing | 8,390 | 7,008 | 1,382 | |

*(Continued)*

**Table 1.** (Continued)

| Characteristic | Overall, N = 13,067 | Screened for cervical cancer | | |
|---|---|---|---|---|
| | | No, N = 10,685 (84.5%) | Yes, N = 2,382 (16.5%) | P values |
| **Wealth quintile (w%)** | | | | <0.001 |
| Lowest | 2,406 (19.2%) | 2,134 (20.6%) | 272 (12.4%) | |
| Second | 2,456 (18.8%) | 2,113 (19.7%) | 343 (14.1%) | |
| Middle | 2,791 (20.7%) | 2,294 (20.8%) | 497 (20.1%) | |
| Fourth | 2,780 (20.8%) | 2,182 (20.2%) | 598 (24.3%) | |
| Highest | 2,632 (20.4%) | 1,961 (18.7%) | 671 (29.1%) | |
| Missing | 2 | 1 | 1 | |
| **Marital status (w%)** | | | | <0.001 |
| Never married | 2,136 (20.1%) | 2,011 (22.9%) | 125 (5.8%) | |
| Married/living together | 7,894 (58.0%) | 6,254 (55.7%) | 1,640 (69.6%) | |
| Divorced/separated | 1,904 (13.7%) | 1,498 (13.2%) | 406 (16.5%) | |
| Widowed | 1,115 (8.2%) | 908 (8.2%) | 207 (8.1%) | |
| Missing | 18 | 14 | 4 | |
| **Access to modern contraceptive methods (w%)** | 5,690 (41.5%) | 4,503 (39.9%) | 1,187 (49.6%) | <0.001 |

CECAP = Cervical Cancer Control Programme; WHO = World Health Organisation.

P values are for Wilcoxon rank-sum test for complex survey samples (continuous variables) and Chi-squared test with Rao & Scott's second-order correction (categorical variables). All Ns reported are unweighted, and all proportions are weighted estimates.

21.3). The Central East zone had the lowest prevalence of self-reported cervical cancer screening in the country (9.4%, 95% CI 7.5–11.2). In all the zones, the majority of the respondents were who reported having undergone screening were from urban regions (22.7% [95% CI 21.4–24.0] versus 15.2% [95% CI 14.0–16.8])–Fig 1C.

For women living with HIV, the prevalence of self-reported screening is highest in Blantyre City (47.6%, 95% CI 40.8–54.3), followed by Lilongwe City (45.8%, 95% CI 35.1–56.5) and the Southwest zone (44.4, 95% CI 38.7–50.0). The lowest reported screening prevalence is observed in the Central East (24.6%, 95% CI 17.0–32.2)–Fig 1B. The screening prevalence distribution for HIV negative women was similar to the overall picture, with the highest prevalence of screening being reported in Blantyre City (21.5%, 95% CI 17.6–25.3), followed by the Southwest (20.1%, 95% CI 17.6–22.6) and Lilongwe City (16.3%, 95% CI 14.7–18.0), and the Central East zone had the lowest prevalence of self-reported cervical cancer screening in the country (8.6%, 95% CI 6.8–10.3).

Fig 2 summarises the prevalence of cervical cancer screening by granulated age group and HIV status. The highest screening prevalence (28.8%) was observed between the ages of 40 to 44 years, followed closely by ages 45 to 49 years at 28.2%. in all age groups, women living with HIV had higher screening prevalence than HIV negative women.

### Factors associated with self-reported cervical cancer screening

Table 2 details the relationships between self-reported cervical cancer screening and the independent variables. Women aged 25 to 49 years had almost three times the odds of reporting having screened for cervical cancer than younger women (adjusted odds ratio (aOR) of 2.86, 95% CI 2.15–3.80)). Women aged 50 and older were 2.90 times (95% CI 2.06–4.10) more likely to report screening than women aged less than 25 years. Women living with HIV were 2.83 times more likely to report screening than HIV negative women (95% CI 2.29–3.50). Women

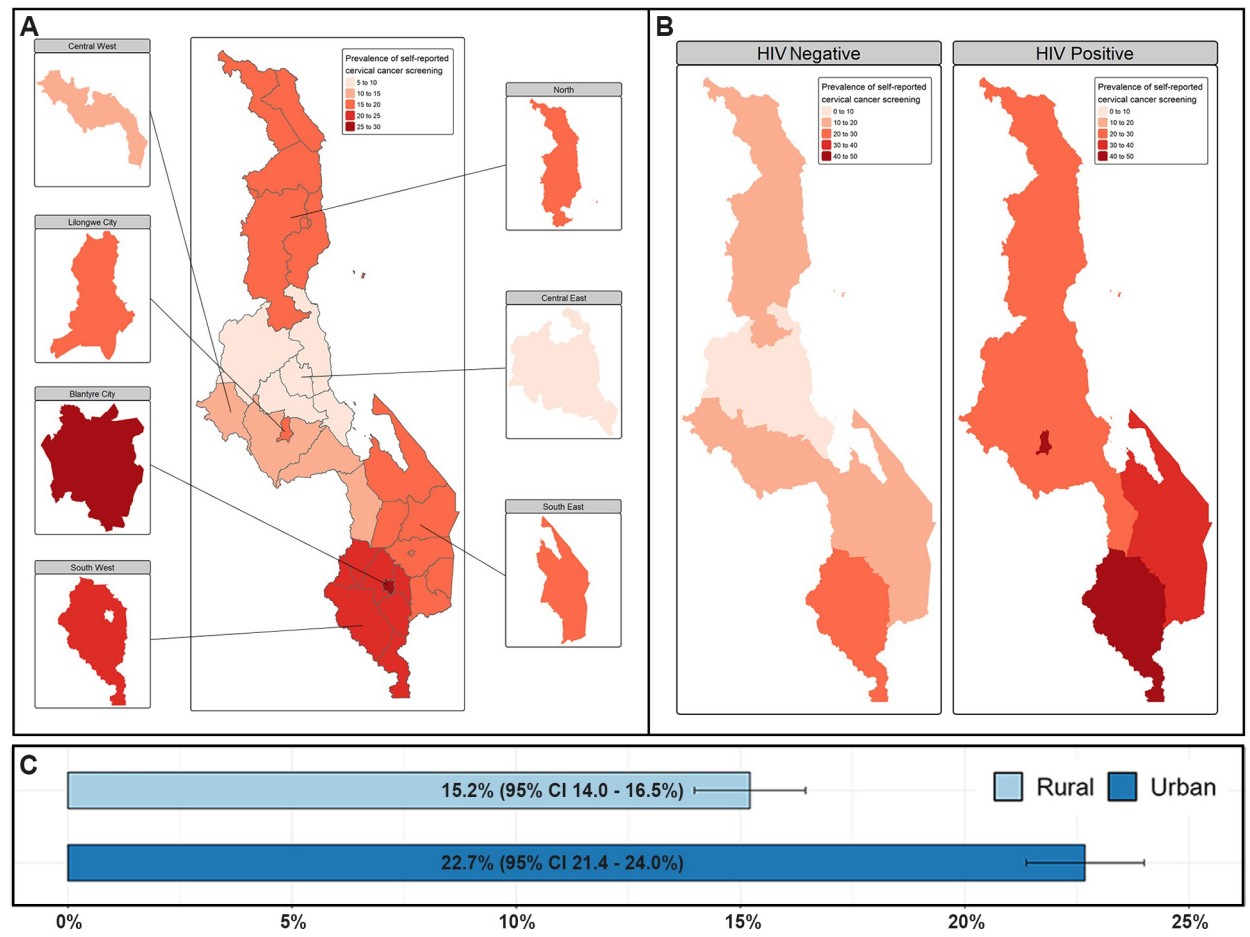

**Fig 1. Geographical distribution of self-reported cervical cancer screening in Malawi.** (A) Overall distribution of cervical cancer screening by region. (B) Regional distribution of cervical cancer screening stratified by HIV status. (C) Prevalence of screening by residential type.

with a higher education level and with the highest wealth quintile were more likely to report having screened compared to the baselines (adjusted odds ratio (aOR) 2.74 [95% CI 1.67–4.52] and aOR 2.86 [95% CI 2.01–4.08] respectively). There was no difference in reported cervical cancer screening between women who reported accessing modern contraceptive methods and those who don't (aOR 1.03, 95% CI 0.86–1.24, p = 0.716).

## Discussion

This study provides an updated nationwide overview of the uptake of cervical cancer screening in Malawi, and describes the spatial distribution and individual-level characteristics associated with cervical cancer screening. We found a low prevalence of reported screening, with screening being more prevalent in the HIV population and in urban settlements. The study also found the prevalence of cervical cancer screening to be lowest in Central East zone of the country, followed by the Central West zone.

The prevalence of cervical cancer screening by self-report in this study was 16.5% (95% CI 15.5–18.0%). It is difficult to make direct comparisons with previous studies due to the lack of comprehensive information on the coverage and uptake of cervical cancer screening in Malawi. The MPHIA 2016 report only reported the prevalence of self-reported cervical cancer

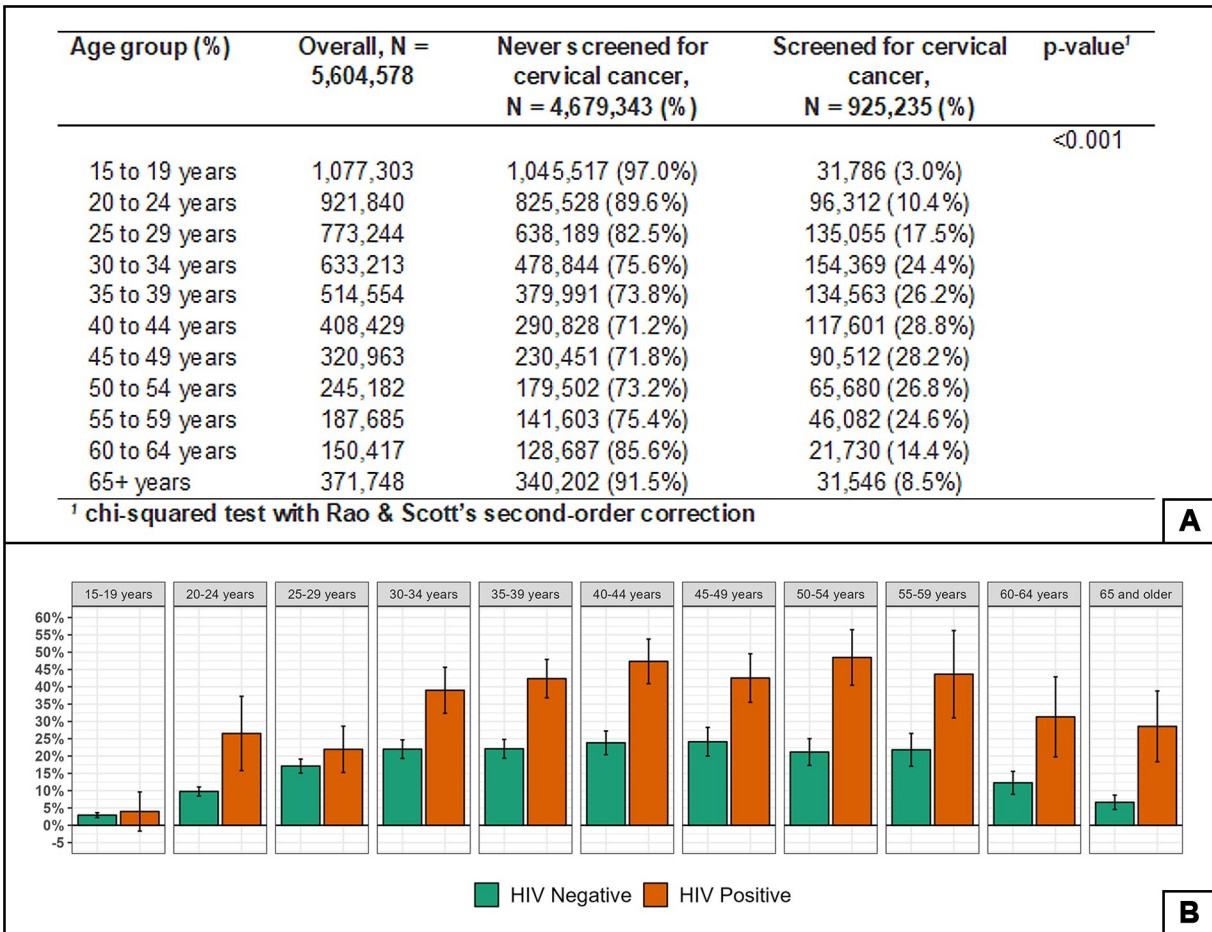

| Age group (%) | Overall, N = 5,604,578 | Never screened for cervical cancer, N = 4,679,343 (%) | Screened for cervical cancer, N = 925,235 (%) | p-value[1] |
|---|---|---|---|---|
| | | | | <0.001 |
| 15 to 19 years | 1,077,303 | 1,045,517 (97.0%) | 31,786 (3.0%) | |
| 20 to 24 years | 921,840 | 825,528 (89.6%) | 96,312 (10.4%) | |
| 25 to 29 years | 773,244 | 638,189 (82.5%) | 135,055 (17.5%) | |
| 30 to 34 years | 633,213 | 478,844 (75.6%) | 154,369 (24.4%) | |
| 35 to 39 years | 514,554 | 379,991 (73.8%) | 134,563 (26.2%) | |
| 40 to 44 years | 408,429 | 290,828 (71.2%) | 117,601 (28.8%) | |
| 45 to 49 years | 320,963 | 230,451 (71.8%) | 90,512 (28.2%) | |
| 50 to 54 years | 245,182 | 179,502 (73.2%) | 65,680 (26.8%) | |
| 55 to 59 years | 187,685 | 141,603 (75.4%) | 46,082 (24.6%) | |
| 60 to 64 years | 150,417 | 128,687 (85.6%) | 21,730 (14.4%) | |
| 65+ years | 371,748 | 340,202 (91.5%) | 31,546 (8.5%) | |
| [1] chi-squared test with Rao & Scott's second-order correction | | | | A |

**Fig 2. Prevalence of self-reported cervical cancer screening.** (A) Prevalence of self-reported cervical cancer screening by age. (B) Prevalence of self-reported cervical cancer screening by HIV status.

screening for the ages of 30 years to 49 years in HIV positive women, which was estimated to be 18.7% [12]. A study by Msyamboza et al. using data from the CECAP database examined the coverage of cervical cancer screening in women aged 30 to 45 years only, which they estimated to be 26.5% in 2015; an increase from the 9.3% reported in 2011 [6]. The prevalence of screening in this age group in our population shows a prevalence of 26.1%, denoting no real change in the coverage of cervical cancer screening.

Cervical cancer screening was more common in women living with HIV than in HIV-negative women. The prevalence of screening in women living with HIV in this study was more than twice the estimated prevalence in HIV negative women. We postulate that this is due to the deliberate efforts to deliver integrated services to people living with HIV by PEPFAR implementing partners such as Elizabeth Glaser Paediatric AIDS Foundation (EGPAF), Johns Hopkins Program for International Education in Gynaecology and Obstetrics (JHPIEGO), Partners in Hope (PIH), and Lighthouse Trust, and other partners such as Médecins Sans Frontières (MSF), including deliberate screening for tuberculosis, enhanced surveillance of sexually transmitted diseases, and scheduled screening for cervical cancer in women [13]. Women who present for refills of their antiretrovirals are routinely referred to the cervical cancer screening clinics in health facilities. Such enhanced referral mechanisms are lacking in

**Table 2. Univariable and multivariable logistics regression describing the factors associated with screening for cervical cancer.**

| Characteristic | cOR | 95% CI | P value | aOR | 95% CI | P value |
|---|---|---|---|---|---|---|
| **Age group based on CECAP targets** | | | <0.001 | | | <0.001 |
| Less than 25 years | — | — | | — | — | |
| 25 to 49 years | 4.57 | 3.93, 5.32 | | 2.86 | 2.15, 3.80 | |
| 50 years and older | 3.05 | 2.55, 3.65 | | 2.90 | 2.06, 4.10 | |
| **HIV status** | | | <0.001 | | | <0.001 |
| Negative | — | — | | — | — | |
| Positive | 3.74 | 3.27, 4.27 | | 2.83 | 2.29, 3.50 | |
| **Ever been pregnant** | | | <0.001 | | | 0.008 |
| No | — | — | | — | — | |
| Yes | 6.55 | 5.17, 8.29 | | 1.93 | 1.19, 3.14 | |
| **Residence type** | | | <0.001 | | | 0.270 |
| Urban | — | — | | — | — | |
| Rural | 0.61 | 0.54, 0.69 | | 1.14 | 0.90, 1.44 | |
| **Education level** | | | <0.001 | | | <0.001 |
| No education | — | — | | — | — | |
| Primary | 1.36 | 1.17, 1.58 | | 1.49 | 1.12, 1.97 | |
| Secondary | 1.66 | 1.41, 1.97 | | 1.72 | 1.25, 2.36 | |
| More than secondary | 3.18 | 2.35, 4.30 | | 2.74 | 1.67, 4.52 | |
| **Occupation** | | | <0.001 | | | 0.102 |
| Unemployed | — | — | | — | — | |
| Mining | 1.00 | 0.21, 4.77 | | 0.71 | 0.12, 4.15 | |
| Transport | 2.92 | 0.87, 9.84 | | 1.77 | 0.54, 5.84 | |
| Construction | 2.29 | 1.26, 4.14 | | 1.84 | 0.98, 3.45 | |
| Uniformed personnel | 3.98 | 2.13, 7.40 | | 2.18 | 1.07, 4.42 | |
| Informal trade | 2.21 | 1.78, 2.75 | | 1.46 | 1.15, 1.85 | |
| Garment industries | 2.45 | 0.94, 6.34 | | 1.26 | 0.49, 3.27 | |
| Housekeeper | 1.45 | 1.05, 1.99 | | 1.27 | 0.88, 1.83 | |
| Student | 0.23 | 0.07, 0.75 | | 0.77 | 0.24, 2.46 | |
| Other | 2.15 | 1.63, 2.84 | | 1.19 | 0.86, 1.65 | |
| **Wealth quintile** | | | <0.001 | | | <0.001 |
| Lowest | — | — | | — | — | |
| Second | 1.19 | 0.98, 1.45 | | 1.24 | 0.93, 1.67 | |
| Middle | 1.61 | 1.29, 2.00 | | 1.71 | 1.25, 2.34 | |
| Fourth | 2.01 | 1.61, 2.51 | | 1.88 | 1.37, 2.58 | |
| Highest | 2.58 | 2.08, 3.20 | | 2.86 | 2.01, 4.08 | |
| **Marital status** | | | <0.001 | | | 0.074 |
| Never married | — | — | | — | — | |
| Married/living together | 4.96 | 4.00, 6.15 | | 1.50 | 0.95, 2.37 | |
| Divorced/separated | 4.96 | 3.88, 6.34 | | 1.41 | 0.83, 2.40 | |
| Widowed | 3.93 | 3.01, 5.14 | | 0.97 | 0.56, 1.67 | |
| **Accesses modern contraceptive methods** | | | <0.001 | | | 0.716 |
| No | — | — | | — | — | |
| Yes | 1.48 | 1.35, 1.63 | | 1.03 | 0.86, 1.24 | |

cOR, Crude Odds Ratio; aOR, Adjusted Odds Ratio; CI, Confidence Interval

most disease and health programs, with postnatal clinics and family planning clinics being the only platform through which HIV negative women receive recommendations for cervical cancer screening, resulting in the strong association between cervical cancer screening and history of pregnancy and access to modern contraceptive methods identified in this study.

The regional HIV prevalence and the activities of partners in the country also inform the distribution of cervical cancer screening uptake in the country. Currently, the MSF, and through PEPFAR funding, EGPAF and AIDS Health Foundation (AHF), have activities in the south of Malawi. These supplement the routine screening facilities set up by the Malawi MoH and the Lighthouse Trust in central hospitals. The central region is served by Baylor and PIH. The PEPFAR funding targets districts with the highest HIV prevalence. As such, districts in the Central and Northern regions, where HIV prevalence is lowest, [11] receive the least partner support for cervical cancer screening.

Regional inequalities in healthcare access across Malawi also exist, with wealth-based disparities in access to maternal healthcare highest in the central region, followed by the north, and lowest in the south, [14] potentially contributing to the observed disparities between the regions. Additionally, the Southern region has a higher number of facilities providing maternal health care services that the Central and Northern regions, [15] though district-level granular data on the number of facilities providing cervical cancer screening services is lacking. Other factors that affect health access may have also contributed to the observed effects between the regions, including geographical terrain, road infrastructure, poverty levels, and community and cultural dynamics [14, 16, 17]. Thus interventions to improve uptake of cervical cancer screening need to incorporate the complex interplay between all the factors potentially contributing to inequities in care access, taking into considerations of the domains of the comprehensive health access framework [18]. Further, cervical cancer screening uptake was found to be higher in urban regions than in rural areas. Differences in healthcare uptake between rural and urban areas can be attributed to several factors. Rural areas are more likely to be impoverished, are at higher risk of social vulnerabilities, and have greater challenges overcoming geographical barriers [17, 19]. Transportation challenges can further hinder screening uptake. A study by Varela et al. demonstrated that transportation was a significant barrier to accessing healthcare in Malawi, and transportation as a challenge was more prevalent for rural residents [20]. Additionally, urban residents tend to have higher socioeconomic status, better health literacy, and more access to educational campaigns about cervical cancer. Moreover, urban areas often benefit more from government and NGO health initiatives due to logistical ease.

Cervical cancer screening was strongly associated with higher education level and higher wealth quintile. Gerstl et al. described both education and income as factors associated with screening for cervical cancer in two southern districts of Malawi [21]. This is also corroborated by a study by Gowokani Chirwa who investigated socioeconomic disparities in cervical cancer uptake using the MPHIA 2015 to 2016 data [22]. It is likely that women who have attained a higher education level and are in a higher income bracket are more aware of cervical cancer and its preventative measures, and allows for overcoming facility barriers such as distance to health facilities, access to health facilities with adequate human, laboratory and clinical resources to conduct screening, and patient navigation challenges [23, 24].

A key facet to preventing cervical cancer is HPV vaccinations. The National Cervical Cancer Control Strategy outlines a target HPV vaccine coverage in girls aged 9–13 years who have of 90% in its strategic framework [8]. The document also detailed challenges to the programme's targets after reviewing the 2004 strategic document, which included the shortfalls in scale-up implementation of the HPV vaccination programme, after the promising results of a pilot vaccine implementation programme in two districts [11, 25]. To our knowledge, up to date studies that evaluate the HPV vaccination programme in Malawi are lacking, suggesting a

knowledge gap that needs to be addressed. Dorji et al., in their meta-analysis, described the uptake of the HPV vaccine in low-income countries including Malawi. They noted that despite some countries reporting high vaccine uptake during reported studies, there is a drastic drop when funding for the particular campaign expires [26]. By the time the MPHIA study was being conducted, the Malawi vaccine programme was only a year old, thus making it difficult to obtain reliable responses about vaccine uptake in the surveyed population. Nonetheless, there remains a large unmet need for vaccination in the target age group in the country, as anecdotal reports outline frequent stockouts of the vaccine, hindering routine provision.

This paper has limitations. Firstly, measurement of the outcome is based on self-reported information, thus there may be a likelihood of recall bias as well as social desirability bias led by societal expectations which may potentially have inflated the prevalence estimate. The reliability of the reported outcome was also vulnerable to effects of response fatigue, especially considering that the MPHIA survey was a relatively lengthy survey. Additionally, due to the self-reported nature of the screening data, we did not have reliable data on treatment and outcomes of screening and treatment, and hence did not report these. Secondly, we could only use the variables that were available in the MPHIA dataset due to the secondary nature of the study. There could be potential confounding in the associations that has not been explored. Despite these, the study recruited a large nationally representative sample of the women in Malawi, allowing for broad and precise generalisations on the uptake of cervical cancer screening in the country.

In conclusion, the uptake of cervical cancer screening in Malawi remains low and is primarily driven by the efforts to address the HIV/AIDS epidemic. Screening efforts are hence concentrated on high HIV-burden, urban regions. Despite this, the uptake of screening remains far below the national target of 80%. There is a need to identify funding support and implementation approaches to potentiate screening coverage in both HIV positive and HIV negative women in the country. Further studies of a longitudinal nature are required in order to describe the complete care cascade, from the proportion of abnormalities identified on screening, to the numbers of women treated for pre-cancerous cervical lesions, and the short- and long-term outcomes of the screen-and-treat services for cervical cancer in Malawi.

## Supporting information

**S1 Fig. Map of Malawi demonstrating the administrative zones.**
(TIF)

**S2 Fig. Proportion of participants who had ever received the HPV vaccination.**
(TIFF)

**S1 Table. Variables included in the analysis.**
(DOCX)

**S2 Table. Variance inflation factors for independent variables considered for the multivariable model.**
(DOCX)

**S3 Table. The proportion of women reporting vaccine uptake by age group.**
(DOCX)

## Author Contributions

**Conceptualization:** Hussein Hassan Twabi, Takondwa Charles Msosa, Marriott Nliwasa.

**Data curation:** Hussein Hassan Twabi.

**Formal analysis:** Hussein Hassan Twabi, Takondwa Charles Msosa.

**Methodology:** Hussein Hassan Twabi.

**Visualization:** Hussein Hassan Twabi.

**Writing – original draft:** Hussein Hassan Twabi.

**Writing – review & editing:** Hussein Hassan Twabi, Takondwa Charles Msosa, Samuel James Meja, Madalo Mukoka, Robina Semphere, Geoffrey Chipungu, David Lissauer, Maria Lisa Odland, Jenny Tudor, Chisomo Msefula, Marriott Nliwasa.

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
