## [Decision Letter · Decision Letter 0]

10 Jun 2024

PONE-D-24-13212Spatial distribution and characteristics of women reporting cervical cancer screening in Malawi: An analysis of nationally representative survey dataPLOS ONE

Dear Dr. Twabi,

Thank you for submitting your manuscript to PLOS ONE. After careful consideration, we feel that it has merit but does not fully meet PLOS ONE’s publication criteria as it currently stands. Therefore, we invite you to submit a revised version of the manuscript that addresses the points raised during the review process.

**1. revise the title as suggested****2. check and revise the biostatistical estimates; significant p-value with overlapping confidence intervals for some variables****3. Address the comment on the final multivariate model****4. issues about self-reported data** 

We look forward to receiving your revised manuscript.

Kind regards,

Jonah Musa, MBBS, MSCI,PhD

Academic Editor

PLOS ONE

Journal Requirements:

[DL is supported by a National Institute for Health and Care Research (NIHR) Global Health Professorship (NIHR300808), using UK aid from the UK Government to support global health research. The views expressed in this publication are those of the authors and not necessarily those of the NIHR or the UK Government.]

 [The author(s) received no specific funding for this work.]

3. Please include your tables as part of your main manuscript and remove the individual files. Please note that supplementary tables (should remain/ be uploaded) as separate ""supporting information"" files".

4. We notice that your supplementary figures are uploaded with the file type 'Figure'. Please amend the file type to 'Supporting Information'. Please ensure that each Supporting Information file has a legend listed in the manuscript after the references list.

Reviewers' comments:

Reviewer's Responses to Questions

**Comments to the Author**

1. Is the manuscript technically sound, and do the data support the conclusions?

Reviewer #1: Partly

Reviewer #2: Yes

2. Has the statistical analysis been performed appropriately and rigorously? 

Reviewer #1: No

Reviewer #2: Yes

3. Have the authors made all data underlying the findings in their manuscript fully available?

Reviewer #1: No

Reviewer #2: Yes

4. Is the manuscript presented in an intelligible fashion and written in standard English?

Reviewer #1: Yes

Reviewer #2: Yes

5. Review Comments to the Author

Reviewer #1: 1. The title should include the survey type like: (Spatial distribution and characteristics of women reporting cervical cancer screening in Malawi: An analysis of Malawi PHIA). You may improve your title by adding the year of the survey.

2. The paper needs to have line numbers for ease of review since it enables referring to line numbers for feedback or comments.

3. The paper reports that there were variations by regions however the between and intra region variations are not provided. This will strengthen your paper.

4. “Since the study by Msyamboza et al. in 2016,6 there has been no updated countrywide estimate of the cervical cancer screening uptake by the general population in Malawi.” but this paper “https://journals.lww.com/ijnc/fulltext/2023/08010/cervical_cancer_care_cascade_among_women_living.7.aspx” focusses on HIV+ women so may be refine this sentence and perhaps cite this paper.

5. The data collection for PHIA is not done in CSPro. I suggest that the authors refer to the report and recast this section.

6. “Descriptive analyses were also made for age categorisations based on the WHO cervical cancer elimination strategy targets, which included age of less than 35 years, age of 35 to 45 years, and age of 46 years and older” should be moved to data analysis section

7. “These zones include the Southeast, Southwest, Central East, Central West, and the Northern zone” should be categorized as North, Centre and South; and if details are needed then a forest plot by district should be plotted.

8. “A total of 12,815 households and 26,519 participants who were 15 years and older were surveyed during the MPHIA survey. There were 22,662 participants who had undergone HIV testing and had HIV test results” should be taken out or go into the methods section.

9. For Table 1: In order not to scare away readers, can you present the un weighted totals and then the weighted percentages. Your current tables are cumbersome and they may often go unread.

10. The authors should not report unadjusted ORs. The focus should be on adjusted effects.

11. The authors need to check the p-value for marital status since the Cis contain 1 but the p-value is significant. Verify the p-value for occupation as well.

12. The statistic on uptake of HPV vaccine is not a correct one if applied to the entire group of women since it was introduced when most of the women of reproductive age may not have had the opportunity as those below 25 years. Perhaps take that out in order not to mis-inform the readership.

13. The authors need to have around 30 references since 18 is too few.

14. How do your results compare with the Malawi WHO STEPS report? This is a key report on NCD and the authors never cited it? Consider 2009 and 2017 STEPS report.

15. One other limitation is that the paper focusses on just those screened due to the data source. Refer to (https://journals.lww.com/ijnc/fulltext/2023/08010/cervical_cancer_care_cascade_among_women_living.7.aspx) for the care cascade information. This could strengthen your paper but also the PHIA project or similar projects to go further by asking the additional questions to ensure a complete care cascade.

16. Overall: What is the knowledge gap that the paper is bridging? I would put this in the first paragraph under discussion so that the paper is more attractive rather than write the paper just as a narrative essay.

17. Overall: How was the final multivariate model arrived at?

Reviewer #2: Overall Review

Strengths:

The major strengths of the Results and Discussion section include clear and logical presentation, appropriate use of statistical methods, comprehensive interpretation of findings, and a thorough discussion of the implications for public health.

The inclusion of well-designed tables and figures enhances the clarity of the presentation.

Weaknesses:

One potential weakness is the reliance on self-reported data, which may introduce bias. This is acknowledged in the limitations but could be further emphasized.

While the discussion is thorough, more detail on regional disparities and the specific barriers in low-screening areas could provide additional valuable insights.

Recommendations:

Emphasize the limitations of self-reported data more strongly and suggest ways to address this in future research.

Provide a more detailed analysis of regional disparities and the specific barriers to cervical cancer screening in low-screening areas.

Overall Recommendation:

Language review

The manuscript is written in standard English and is generally clear and intelligible. However, there are some areas where improvements could be made to enhance readability and correctness. Here are some specific observations and suggestions:

Abstract:

• The abstract provides a concise summary of the study. Still, it has a minor typographical error: "Malawi has one of the is a country with the highest incidence and mortality rates of cervical cancer in the world." This should be corrected to: "Malawi has one of the highest incidence and mortality rates of cervical cancer in the world."

Main Text

Introduction:

• The introduction is well-written and sets the context for the study. However, some sentences could be rephrased for clarity:

o "Malawi has one of the is a country with the highest incidence and mortality rates of cervical cancer in the world." should be corrected as mentioned above.

o "Despite a national strategic plan to eliminate cervical cancer, as well as a national roll-out of VIA and screen-and-treat services in Malawi, coverage remains far below the national target." This sentence is clear but could be more concise: "Despite a national strategic plan and the roll-out of VIA and screen-and-treat services, cervical cancer screening coverage in Malawi remains far below the national target."

Methods:

• The methods section is detailed and clearly describes the study design, data collection, and statistical analysis. A few minor adjustments for clarity:

o "The survey, carried out between January 2020 and April 2021." This fragment should be completed or integrated into the previous sentence for better flow.

o "We used univariable and multivariable logistics regression approaches to examine associations..." should be "logistic regression" instead of "logistics regression."

Results:

• The results are presented clearly, with appropriate use of tables and figures. Some sentences could be refined for clarity:

o "The overall prevalence of self-reported cervical cancer screening was 16.5% (95% CI 15.5–18.0%). The prevalence of screening was 37.8% (95% CI 34.8–40.9) in women living with HIV and 14.0% (95% CI 13.0–15.0) in HIV negative women." This could be rephrased for better readability: "The overall prevalence of self-reported cervical cancer screening was 16.5% (95% CI 15.5–18.0%). Among women living with HIV, the prevalence was 37.8% (95% CI 34.8–40.9), compared to 14.0% (95% CI 13.0–15.0) in HIV-negative women."

Discussion:

• The discussion appropriately interprets the results and places them in the context of existing literature. Some sentences could be simplified:

o "Screening for cervical cancer was predominant in women living with HIV as compared to the HIV negative population." could be simplified to: "Cervical cancer screening was more common in women living with HIV than in HIV-negative women."

Conclusion:

• The conclusion is clear and summarizes the main findings effectively. Minor grammatical improvements could be made:

o "The uptake of cervical cancer screening remains low in the country and is largely driven by the efforts to address the HIV/AIDS epidemic." This could be rephrased to: "The uptake of cervical cancer screening in Malawi remains low and is primarily driven by efforts to address the HIV/AIDS epidemic."

Specific Typographical and Grammatical Errors

1. Typographical Errors:

o "Malawi has one of the is a country with the highest incidence and mortality rates of cervical cancer in the world." should be corrected to: "Malawi has one of the highest incidence and mortality rates of cervical cancer in the world."

o "logistics regression" should be "logistic regression."

2. Grammatical Errors:

o "The survey, carried out between January 2020 and April 2021." should be integrated into a complete sentence.

o "Screening for cervical cancer was predominant in women living with HIV as compared to the HIV negative population." can be simplified for clarity.

Conclusion

I recommend this manuscript for publication with minor revisions. The suggested revisions include emphasizing the limitations of self-reported data and providing a more detailed analysis of regional disparities in cervical cancer screening rates. These revisions will enhance the clarity and comprehensiveness of the manuscript, making it a valuable contribution to the field of public health.

6. PLOS authors have the option to publish the peer review history of their article (what does this mean?). If published, this will include your full peer review and any attached files.

Reviewer #1: No

Reviewer #2: **Yes: **Mansoor Farahani

---

## [Author Response · Author response to Decision Letter 0]

18 Jul 2024

Responses to editorial comments

Please ensure that your manuscript meets PLOS ONE's style requirements, including those for file naming. The PLOS ONE style templates can be found at https://journals.plos.org/plosone/s/file?id=wjVg/PLOSOne_formatting_sample_main_body.pdf and https://journals.plos.org/plosone/s/file?id=ba62/PLOSOne_formatting_sample_title_authors_affiliations.pdf. 

Response: The manuscript has been amended to fit the style requirements

2. Thank you for stating the following in the Acknowledgments Section of your manuscript: [DL is supported by a National Institute for Health and Care Research (NIHR) Global Health Professorship (NIHR300808), using UK aid from the UK Government to support global health research. The views expressed in this publication are those of the authors and not necessarily those of the NIHR or the UK Government.]

We note that you have provided funding information that is not currently declared in your Funding Statement. However, funding information should not appear in the Acknowledgments section or other areas of your manuscript. We will only publish funding information present in the Funding Statement section of the online submission form. Please remove any funding-related text from the manuscript and let us know how you would like to update your Funding Statement. Currently, your Funding Statement reads as follows: [The author(s) received no specific funding for this work.]

Response: The funding statement should be amended as follows: [None of the authors received specific funding for this work]. The acknowledgement section has been removed from the manuscript

3. Please include your tables as part of your main manuscript and remove the individual files. Please note that supplementary tables (should remain/ be uploaded) as separate ""supporting information"" files".

Response: Tables have been included in the manuscript body

4. We notice that your supplementary figures are uploaded with the file type 'Figure'. Please amend the file type to 'Supporting Information'. Please ensure that each Supporting Information file has a legend listed in the manuscript after the references list.

Response: A list of legends for the Supporting Information files has been included after the references list and all supporting documentation have been uploaded appropriately

Response: Captions and citations have been included as suggested

Responses to Reviewer 1 comments

1. The title should include the survey type like: (Spatial distribution and characteristics of women reporting cervical cancer screening in Malawi: An analysis of Malawi PHIA). You may improve your title by adding the year of the survey.

Response: the title has been amended as suggested 

2. The paper needs to have line numbers for ease of review since it enables referring to line numbers for feedback or comments.

Response: Line numbers have been included to the manuscript 

3. The paper reports that there were variations by regions however the between and intra region variations are not provided. This will strengthen your paper.

Response: The discussion has been expanded to address this comment

4. “Since the study by Msyamboza et al. in 2016,6 there has been no updated countrywide estimate of the cervical cancer screening uptake by the general population in Malawi.” but this paper “https://journals.lww.com/ijnc/fulltext/2023/08010/cervical_cancer_care_cascade_among_women_living.7.aspx” focusses on HIV+ women so may be refine this sentence and perhaps cite this paper.

Response: The statement beginning “Since the study ….” Is referring to studies in the general population (encompassing both HIV positive and HIV negative populations) and thus excluded this study. The preceding paragraph has been updated to include a statement about the study in HIV positive women and includes the suggested paper as a reference (Line 107 to 112). 

5. The data collection for PHIA is not done in CSPro. I suggest that the authors refer to the report and recast this section.

Response: The MPHIA 2020-2021 report states in the Survey Design, Methods, And Response Rates section that “Questionnaire and field laboratory data were collected on mobile tablet devices using an application programmed in Census and Survey Processing System (CSPro) software, an open-source mobile data collection application.” [Ministry of Health (MOH), Malawi. Malawi Population-based HIV Impact Assessment 2020-2021 (MPHIA 2020-2021): Final Report. Lilongwe: Ministry of Health, Malawi; 2022 Nov.]

6. “Descriptive analyses were also made for age categorizations based on the WHO cervical cancer elimination strategy targets, which included age of less than 35 years, age of 35 to 45 years, and age of 46 years and older” should be moved to data analysis section

Response: The statement has been amended to “Age categorizations based were also made based on the WHO cervical cancer elimination strategy targets, which included age of less than 35 years, age of 35 to 45 years, and age of 46 years and older” and has been left under the Variable Definitions section as it now fits better there. 

7. “These zones include the Southeast, Southwest, Central East, Central West, and the Northern zone” should be categorized as North, Centre and South; and if details are needed then a forest plot by district should be plotted. 

Response: The zone categorization adopted in this paper are standard zones that are used in Health administration in Malawi and are as categorized in the MPHIA survey (as well as the WHO STEPS survey). This categorization allows for a more granular view of the uptake in the country in the absence of district level data. The homogeneity between this sub regions is evident in the uptake of screening between the Southwest and Southeast, and the Central West and Central East. [see Figure 1] District level data was not available in the provided dataset as the survey only collected centroid coordinates and not household coordinates to ensure the anonymity of the respondents was not jeopardized. 

8. “A total of 12,815 households and 26,519 participants who were 15 years and older were surveyed during the MPHIA survey. There were 22,662 participants who had undergone HIV testing and had HIV test results” should be taken out or go into the methods section.

Response: This section of the results addresses the “Participants” sub-section as outlined in the STROBE guidelines. As such we believe it best placed under the Results section in line with the standard reporting guidelines. (https://www.strobe-statement.org/download/strobe-checklist-cohort-case-control-and-cross-sectional-studies-combined) 

9. For Table 1: In order not to scare away readers, can you present the un weighted totals and then the weighted percentages. Your current tables are cumbersome and they may often go unread.

Response: Table 1 has been amended as recommended

10. The authors should not report unadjusted ORs. The focus should be on adjusted effects.

Response: The results have been amended as recommended

11. The authors need to check the p-value for marital status since the Cis contain 1 but the p-value is significant. Verify the p-value for occupation as well.

Response: The results have been reviewed and an error was indeed made in transcribing the p value. This has been corrected. It is worth noting that the Global P does not represent the significance of individual strata for each variable in the multivariable model, but rather the significance of the variable in the model itself. 

12. The statistic on uptake of HPV vaccine is not a correct one if applied to the entire group of women since it was introduced when most of the women of reproductive age may not have had the opportunity as those below 25 years. Perhaps take that out in order not to mis-inform the readership.

Response: The section on vaccine uptake has been removed, and the discussion has been amended to clarify this. A supplementary figure and table have been included demonstrating the proportion of vaccine uptake by age group. 

13. The authors need to have around 30 references since 18 is too few.

Response: More references have been cited with the expansion of the discussions section

14. How do your results compare with the Malawi WHO STEPS report? This is a key report on NCD and the authors never cited it? Consider 2009 and 2017 STEPS report.

Response: We did not find explicit mention of cervical cancer screening uptake in both the 2009 and 2017 STEPS survey publications. Our paper only provides background on cervical cancer. Even so, both STEPS papers do not provide explicit information with regards to cervical cancer screening in Malawi. Despite collecting district data, the reports are also lacking any discussion on patterns of NCDs by districts/regions in Malawi. 

15. One other limitation is that the paper focusses on just those screened due to the data source. Refer to (https://journals.lww.com/ijnc/fulltext/2023/08010/cervical_cancer_care_cascade_among_women_living.7.aspx) for the care cascade information. This could strengthen your paper but also the PHIA project or similar projects to go further by asking the additional questions to ensure a complete care cascade.

Response: This has been noted and a statement has been included in the conclusions to allude to this

16. Overall: What is the knowledge gap that the paper is bridging? I would put this in the first paragraph under discussion so that the paper is more attractive rather than write the paper just as a narrative essay.

Response: Our writing approach follows the LSHTM style of writing the discussion, starting with reiterating the objective of the paper, then reporting the key findings. We have included a statement that provides the impact of the paper, in line with your suggestion.

17. Overall: How was the final multivariate model arrived at?

Response: The multivariable model is an exploratory one, i.e. it begins to provide some picture of factors that may be associated with cervical cancer screening uptake. It is not causal model, and thus does not follow standard causal model building approaches. The variables were identified a priori as potential factors that may affect uptake of screening. Geographical zone was dropped as it demonstrated a high Variance inflation factor, denoting multicollinearity in the model. Some explanation of this was provided in the methods section.

Responses to Reviewer 2 comments

One potential weakness is the reliance on self-reported data, which may introduce bias. This is acknowledged in the limitations but could be further emphasized.

Response: The statement on this limitation has been expanded to address this comment

While the discussion is thorough, more detail on regional disparities and the specific barriers in low-screening areas could provide additional valuable insights.

Response: A paragraph on other factors that could contribute to the perceived disparities has been included to the discussion

Language review

The manuscript is written in standard English and is generally clear and intelligible. However, there are some areas where improvements could be made to enhance readability and correctness. Here are some specific observations and suggestions:

Abstract:

• The abstract provides a concise summary of the study. Still, it has a minor typographical error: "Malawi has one of the is a country with the highest incidence and mortality rates of cervical cancer in the world." This should be corrected to: "Malawi has one of the highest incidence and mortality rates of cervical cancer in the world."

Response: The sentence has been corrected

Main Text

Introduction:

• The introduction is well-written and sets the context for the study. However, some sentences could be rephrased for clarity:

o "Malawi has one of the is a country with the highest incidence and mortality rates of cervical cancer in the world." should be corrected as mentioned above.

Response: This has been addressed as suggested

o "Despite a national strategic plan to eliminate cervical cancer, as well as a national roll-out of VIA and screen-and-treat services in Malawi, coverage remains far below the national target." This sentence is clear but could be more concise: "Despite a national strategic plan and the roll-out of VIA and screen-and-treat services, cervical cancer screening coverage in Malawi remains far below the national target."

Response: The sentence has been corrected

Methods:

• The methods section is detailed and clearly describes the study design, data collection, and statistical analysis. A few minor adjustments for clarity:

o "The survey, carried out between January 2020 and April 2021." This fragment should be completed or integrated into the previous sentence for better flow.

Response: The sentence was amended

o "We used univariable and multivariable logistics regression approaches to examine associations..." should be "logistic regression" instead of "logistics regression."

Response: The sentence was amended

Results:

• The results are presented clearly, with appropriate use of tables and figures. Some sentences could be refined for clarity:

o "The overall prevalence of self-reported cervical cancer screening was 16.5% (95% CI 15.5–18.0%). The prevalence of screening was 37.8% (95% CI 34.8–40.9) in women living with HIV and 14.0% (95% CI 13.0–15.0) in HIV negative women." This could be rephrased for better readability: "The overall prevalence of self-reported cervical cancer screening was 16.5% (95% CI 15.5–18.0%). Among women living with HIV, the prevalence was 37.8% (95% CI 34.8–40.9), compared to 14.0% (95% CI 13.0–15.0) in HIV-negative women."

Response: The sentence was amended as recommended

Discussion:

• The discussion appropriately interprets the results and places them in the context of existing literature. Some sentences could be simplified:

o "Screening for cervical cancer was predominant in women living with HIV as compared to the HIV negative population." could be simplified to: "Cervical cancer screening was more common in women living with HIV than in HIV-negative women."

Response: The sentence was amended as recommended

Conclusion:

• The conclusion is clear and summarizes the main findings effectively. Minor grammatical improvements could be made:

o "The uptake of cervical cancer screening remains low in the country and is largely driven by the efforts to address the HIV/AIDS epidemic." This could be rephrased to: "The uptake of cervical cancer screening in Malawi remains low and is primarily driven by efforts to address the HIV/AIDS epidemic."

Response: The sentence was amended as recommended

Specific Typographical and Grammatical Errors

1. Typographical Errors:

o "Malawi has one of the is a country with the highest incidence and mortality rates of cervical cancer in the world." should be corrected to: "Malawi has one of the highest incidence and mortality rates of cervical cancer in the world."

Response: The sentence was amended as recommended

o "logistics regression" should be "logistic regression."

Response: The sentence was amended as recommended

2. Grammatical Errors:

o "The survey, carried out between January 2020 and April 2021." should be integrated into a complete sentence.

Response: The sentence was amended as recommended

o "Screening for cervical cancer was predominant in women living with HIV as compared to the HIV negative population." can be simplified for clarity.

Response: The sentence was amended as recommended

---

## [Decision Letter · Decision Letter 1]

6 Aug 2024

Spatial distribution and characteristics of women reporting cervical cancer screening in Malawi: An analysis of the 2020 to 2021 Malawi Population-based HIV Impact Assessment survey data

PONE-D-24-13212R1

Dear Dr. Twabi,

We’re pleased to inform you that your manuscript has been judged scientifically suitable for publication and will be formally accepted for publication once it meets all outstanding technical requirements.

Kind regards,

Jonah Musa, MBBS, MSCI,PhD

Academic Editor

PLOS ONE

Additional Editor Comments (optional):

Reviewers' comments:

Reviewer's Responses to Questions

**Comments to the Author**

1. If the authors have adequately addressed your comments raised in a previous round of review and you feel that this manuscript is now acceptable for publication, you may indicate that here to bypass the “Comments to the Author” section, enter your conflict of interest statement in the “Confidential to Editor” section, and submit your "Accept" recommendation.

Reviewer #1: All comments have been addressed

2. Is the manuscript technically sound, and do the data support the conclusions?

Reviewer #1: Yes

3. Has the statistical analysis been performed appropriately and rigorously? 

Reviewer #1: Yes

4. Have the authors made all data underlying the findings in their manuscript fully available?

Reviewer #1: Yes

5. Is the manuscript presented in an intelligible fashion and written in standard English?

Reviewer #1: Yes

6. Review Comments to the Author

Reviewer #1: The authors have addressed all the comments. The paper can be published. All ethical issues have been addressed. The authors should consider making available the code chunks to enhance replication.

7. PLOS authors have the option to publish the peer review history of their article (what does this mean?). If published, this will include your full peer review and any attached files.

Reviewer #1: No

---

## [Editor Report · Acceptance letter]

14 Aug 2024

PONE-D-24-13212R1 

PLOS ONE

Dear Dr. Twabi, 

I'm pleased to inform you that your manuscript has been deemed suitable for publication in PLOS ONE. Congratulations! Your manuscript is now being handed over to our production team.

Kind regards, 

on behalf of

Dr. Jonah Musa 

Academic Editor

PLOS ONE